# Factors associated with COVID-19 vaccine uptake among people with type 2 diabetes in Kenya and Tanzania: a mixed-methods study

Peter Binyaruka [1], Sally M Mtenga,[1] Irene Mashasi,[1] Caroline H Karugu [2], Shukri F Mohamed [2], Gershim Asiki,[2] Frances S Mair [3], Cindy M Gray [3,4]

PB and SMM are joint first authors.

¹Department of Health System, Impact Evaluation and Policy, Ifakara Health Institute, Dar es Salaam, Tanzania
²Chronic Disease Management Unit, African Population and Health Research Center, Nairobi, Kenya
³School of Health and Wellbeing, University of Glasgow, Glasgow, UK
⁴School of Social and Political Sciences, University of Glasgow, Glasgow, UK

**Correspondence to**
Dr Peter Binyaruka;
pbinyaruka@ihi.or.tz

## ABSTRACT

**Background** People with type 2 diabetes (T2D) are at increased risk of poor outcomes from COVID-19. Vaccination can improve outcomes, but vaccine hesitancy remains a major challenge. We examined factors influencing COVID-19 vaccine uptake among people with T2D in two sub-Saharan Africa countries that adopted different national approaches to combat COVID-19, Kenya and Tanzania.

**Methods** A mixed-methods study was conducted in February-March 2022, involving a survey of 1000 adults with T2D (500 Kenya; 500 Tanzania) and 51 in-depth interviews (21 Kenya; 30 Tanzania). Determinants of COVID-19 vaccine uptake were identified using a multivariate logistic regression model, while thematic content analysis explored barriers and facilitators.

**Results** COVID-19 vaccine uptake was lower in Tanzania (26%) than in Kenya (75%), which may reflect an initial political hesitancy about vaccines in Tanzania. People with college/university education were four times more likely to be vaccinated than those with no education (Kenya AOR=4.25 (95% CI 1.00 to 18.03), Tanzania AOR=4.07 (1.03 to 16.12)); and people with health insurance were almost twice as likely to be vaccinated than those without health insurance (Kenya AOR=1.70 (1.07 to 2.70), Tanzania AOR=1.81 (1.04 to 3.13)). Vaccine uptake was higher in older people in Kenya, and among those with more comorbidities and higher socioeconomic status in Tanzania. Interviewees reported that wanting protection from severe illness promoted vaccine uptake, while conflicting information, misinformation and fear of side-effects limited uptake.

**Conclusion** COVID-19 vaccine uptake among people with T2D was suboptimal, particularly in Tanzania, where initial political hesitancy had a negative impact. Policy-makers must develop strategies to reduce fear and misconceptions, especially among those who are less educated, uninsured and younger.

## INTRODUCTION

The COVID-19 pandemic has negatively impacted social, health and economic welfare globally. In the health sector, COVID-19 led to healthcare disruptions affecting routine care, especially for patients living with chronic or non-communicable diseases like type 2 diabetes (T2D).[1] People with T2D have a compromised immune system and are at higher risk of infection and adverse outcomes from COVID-19.[2 3]

Vaccination is a proven pharmaceutical intervention to reduce COVID-19 transmission and adverse outcomes.[4] In December 2020, newly developed COVID-19 vaccines were authorised, and efforts by the COVID-19 Vaccines Global Access (COVAX) initiative aimed to ensure global vaccine equity. However, vaccination coverage is still suboptimal in many low-/middle-income countries (LMICs) due to limited COVAX vaccine supply[5] and vaccine hesitancy because of concerns about life-threatening side effects, risk of new diseases and infertility.[6–10]

High vaccine coverage is required to reduce incidence of COVID-19 and hospitalisation. In many countries, people with chronic diseases, such as T2D, are prioritised during national COVID-19 vaccine rollouts.[11] It is therefore important for policy-makers to understand the prevalence of COVID-19 vaccine uptake and factors influencing uptake among people with chronic diseases.

**STRENGTHS AND LIMITATIONS OF THIS STUDY**

⇒ This is the first large population-based study to examine COVID-19 vaccine uptake among people with type 2 diabetes (T2D) in two lower middle-income countries in sub-Saharan Africa.
⇒ The design allowed an exploration of the factors, including political factors, that influenced vaccine uptake in two countries with different approaches to combating COVID-19.
⇒ The mixed-methods study design provided in-depth contextual insights of people's attitudes to and experiences of COVID-19 vaccination.
⇒ A limitation was that COVID-19 vaccination status was based on self-report rather than clinical records.

A few studies among people with T2D, including only two from sub-Saharan Africa, revealed that vaccine uptake was significantly higher among men, public servants, urban populations and people with a higher education level.[12–20] Our paper extends the evidence base by estimating the prevalence of COVID-19 vaccine uptake among people with T2D in Kenya and Tanzania, and identifying the factors influencing uptake in both countries. We chose these two countries because they took different approaches to combat COVID-19 (eg, 2020 lockdown in Kenya, but not in Tanzania; initial government hesitancy about COVID-19 vaccines in Tanzania, but not in Kenya). We focused on T2D because it is a risk factor for other chronic illness such as kidney failure, cardiovascular disease and hypertension,[21] and increases vulnerability and severity of COVID-19.[2 3] Our findings will guide targeted communication campaigns and other strategies to improve vaccination coverage, especially among people with chronic diseases, such as T2D.

## METHODS
### Study setting
This study was conducted in two LMICs in East Africa, Kenya and Tanzania. We purposively selected 4 counties (2 urban—Nairobi, Kiambu; 2 rural—Vihiga, Nyeri) out of 47 counties in Kenya, and 2 regions (one urban—Dar es Salaam; one rural/ peri-urban—Morogoro) out of 31 regions in Tanzania. The prevalence of diabetes mellitus was relatively lower in Kenya (2.4% in 2015)[22] than in Tanzania (9.1% in 2012).[23] By August 2023, Tanzania and Kenya had recorded 43 078 and 343 918 confirmed cases and 846 and 5689 COVID-19-related deaths, respectively.[24] COVID-19 vaccination is free in both countries, but priority was initially given to health workers, older and people with chronic health conditions. Both countries received vaccines through COVAX, but the vaccination rollout began slightly earlier in Kenya (March 2021)[25] compared with Tanzania (July 2021).[26]

### Study design
We use a mixed-methods study design involving a cross-sectional survey of 500 people with T2D in each country, and qualitative in-depth interviews with 21 and 30 people with T2D in Kenya and Tanzania, respectively.

### Study participants and sampling
The survey included adults (18+ years) who were diagnosed with T2D before COVID-19 (ie, before March 2020) in Nairobi (n=276), Kiambu (n=104), Vihiga (n=76), and Nyeri (n=44) counties in Kenya, and in Dar es Salaam (n=300) and Morogoro (n=200) regions in Tanzania. The sample size of 500 people was estimated using the Cochran formula, assuming 50% of patients with T2D experienced disruption of care during COVID-19,[27] with 80% power, a 5% margin of error and a 30% non-response rate. Adult patients with T2D who received care in selected hospitals and health centres before March

2020 were identified from health facility outpatient registers and approached for written informed consent. Based on convenience sampling, participants for the in-depth interviews were mainly selected from survey participants. In both countries, data collection stopped once saturation of views was achieved. People with T2D who were vaccinated or unvaccinated, and with or without other health conditions were eligible to take part in both the survey and in-depth interviews.

### Data collection and recruitment
Fieldworkers were recruited from each country and trained for 4 days before data collection. Trained fieldworkers administered structured survey questionnaires in both countries and used an interview topic guide (online supplemental file 1) to explore facilitators and barriers to COVID-19 vaccination. Both tools were piloted and refined before use. Data collection was conducted in February–March 2022. In Tanzania, all data collection was conducted face to face at the health facility in a room or other location that ensured full privacy for the participant. In Kenya, the survey was administered by phone due to national guidelines to minimise COVID-19 transmission, while the decision was taken to conduct the qualitative interviews face-to-face as they involved a much smaller sample size, and thus lower risk. Data collection was conducted in Swahili in Tanzania, while in Kenya, where fieldworkers were fluent in Swahili and English, participants were allowed to respond in either language.

### Measures
Our primary outcome of interest was COVID-19 vaccine uptake among people with T2D in Kenya and Tanzania and was assessed using the following question: *'Have you been vaccinated for COVID-19?'*. We used a binary outcome with a value of 1 if a participant confirmed having received any COVID-19 vaccine (regardless of dosage) by February 2022, and 0 otherwise.

The explanatory variables as potential determinants of COVID-19 vaccine uptake were based on the WHO's social determinants of health conceptual framework[28] and relevant empirical evidence on factors influencing healthcare seeking behaviour, particularly vaccination.[29] We used two categories of determinants: sociodemographic and health related.

*Sociodemographic factors* included five binary variables: place of residence (rural/urban), sex (male/female), marital status (married/not married), health insurance (insured/non-insured), and socioeconomic status (SES) based on a ladder scale (1–10) that asked participants to rank their households in terms of economic position, where 1 was lowest economic position and 10 was highest economic position (lower 1–5/higher 6–10). We also included four categorical variables: age (<40/40–49/50–59/60–69/>70 years), education level (no education/primary/secondary/higher education), religion (Catholic/Protestant/Muslim), and occupation status

(formal sector workers/farmers/self-employed/retired/unemployed).

*Health-related factors* included three binary variables: family history of T2D (history/no history), time living with T2D (<6 years/≥6 years), and presence of comorbidities (with/without comorbidities). The number of comorbidities was assessed by asking participants to report additional long-term conditions.

## Data analysis

Descriptive analysis was undertaken to describe participants' sociodemographic and health-related characteristics and prevalence of COVID-19 vaccination in each country. To identify the independent determinants of vaccine uptake, we performed separate multivariate logistic regressions for each country. Since vaccine uptake was higher in Kenya (75%—reflecting a non-rare event), we checked the sensitivity of the logistic regression by applying a modified Poisson regression model.[30] We assessed multicollinearity between independent variables using a pairwise correlation analysis. All correlation coefficients were below 0.5, which is considered moderate, supporting the inclusion of all variables in the regression analysis. STATA V.16 was used to analyse the quantitative data.

Audio-recorded data from the qualitative interviews were transcribed verbatim in Kenya and Tanzania. Researchers repeatedly reviewed the transcripts to familiarise themselves with the data. Thematic content analysis was employed to identify common phrases related to the factors that facilitated and deterred COVID-19 vaccine uptake. In this process, inductive coding was used independently by researchers from each study country to categorise the key themes. They then met to discuss and agree on the codes that were applied to all transcripts. NVivo software (V.10 in Kenya, V.12 Tanzania) was used to manage and analyse the data.

## Patient and public involvement

Patients and/or the public were not involved in the design, or conduct, or reporting, or dissemination plans of this research.

## RESULTS

### Participant characteristics

Most participants in both countries were from urban settings (>59%), female (>65%), and married (>62%) (table 1). The majority were also aged above 50 years (the average age was 58 years in Kenya and 57 years in Tanzania) and had completed primary or secondary education. However, religious affiliation differed between the two countries, with almost equal numbers of Muslims and Christians in Tanzania, while Christians dominated in Kenya. In both countries, the majority of participants worked as farmers or were self-employed, had health insurance (>60%) and were of lower SES (89% in Kenya, 75% in Tanzania). Fewer participants in Tanzania (42%)

had a family history of T2D than in Kenya (53%). Most patients in both countries had lived with T2D for 6 years or more (>61%), while the mean number of comorbidities was 3.4 in Kenya and 1.5 in Tanzania.

Most people who participated in the qualitative in-depth interviews had comorbidities (particularly hypertension). They were aged between 60 and 80 years in Kenya, and between 46 and 51 years in Tanzania. In urban areas, interviewees were employed/running businesses, retired or unemployed, while in rural areas the majority were farmers.

### Factors influencing COVID-19 vaccine uptake

The survey data showed that COVID-19 vaccine uptake was relatively lower in Tanzania than in Kenya. Only 26% of participants in Tanzania reported having received at least one COVID-19 vaccination, compared with 75% in Kenya. The qualitative interviews suggested that the low rate of vaccine uptake in Tanzania may reflect the political approach to COVID-19 in the country. Initially, the past presidential regime promoted natural remedies like lemon, ginger and steam with herbal remedies to combat COVID-19, as well as WHO recommended prevention measures (eg, masks and hands sanitizer), but did not endorse COVID-19 vaccination. Despite the current Tanzanian government's support and promotion of COVID-19 vaccination rollout from July 2021, the initial political hesitancy emerged as a continued barrier to vaccine uptake in the accounts of Tanzanian interviewees.

> I cannot say the COVID-19 vaccine is good because our late President said the vaccine cannot protect us from COVID-19 rather we should protect ourselves and said absolutely there is no vaccine to cure COVID-19, so we are following him [Female-PT09-urban-Tanzania].

The survey data showed that participants with a college/university education were over four times more likely to be vaccinated than those with no education in Kenya (AOR=4.25, 95% CI 1.00 to 18.03, p<0.05) and Tanzania (AOR=4.07, 95% CI 1.03 to 16.12, p<0.05) (table 2). The qualitative interviews revealed the importance of education, particularly in terms of understanding and weighing up the complex information available about COVID-19. For instance, some patients who were vaccinated in both countries felt that protecting themselves from the risk of severe illness outweighed any perceived risk from the COVID-19 vaccine.

> I just think it is good. I took AstraZeneca and it did not affect me in any way. In fact, it gave me a sense of safety. You feel you are a little bit safe [Female-220312_1439-rural-Kenya]

However, some interviewees also described the difficulty people faced in trying to navigate the diverse and sometimes conflicting information from different sources, such as government leaders, religious leaders and healthcare professionals. Some participants reported

Table 1 Sociodemographic and health-related characteristics of patients with type 2 diabetes (T2D)

| Variable | Description | Kenya (N=500) | | Tanzania (N=500) | |
|---|---|---|---|---|---|
| | | N | % | N | % |
| Place of residence | Urban | 380 | 76.0% | 300 | 60.0% |
| | Rural/ peri-urban | 120 | 24.0% | 200 | 40.0% |
| Sex | Female | 330 | 66.0% | 336 | 67.2% |
| | Male | 170 | 34.0% | 164 | 32.8% |
| Marital status | Married | 319 | 63.8% | 312 | 62.4% |
| | Not married | 181 | 36.2% | 188 | 37.6% |
| Age group | <40 years | 39 | 7.8% | 36 | 7.2% |
| | 40–49 years | 80 | 16.0% | 62 | 12.4% |
| | 50–59 years | 135 | 27.0% | 165 | 33.0% |
| | 60–69 years | 155 | 31.0% | 206 | 41.2% |
| | ≥70 years | 91 | 18.2% | 31 | 6.2% |
| Mean age in years (SD) | | 500 | 58.2 (12.6) | 500 | 56.8 (10.2) |
| Education level | No education | 16 | 3.2% | 34 | 6.8% |
| | Primary education | 208 | 41.6% | 298 | 59.6% |
| | Secondary education | 201 | 40.2% | 126 | 25.2% |
| | Higher education | 75 | 15.0% | 42 | 8.4% |
| Religion | Catholic | 104 | 20.8% | 137 | 27.4% |
| | Protestants | 379 | 75.8% | 110 | 22.0% |
| | Muslims | 17 | 3.4% | 253 | 50.6% |
| Occupation | Formal sector workers | 31 | 6.2% | 40 | 8.0% |
| | Farmers (small/large scale) | 78 | 15.6% | 103 | 20.6% |
| | Self-employed (small/large business) | 162 | 32.4% | 190 | 38.0% |
| | Retired | 55 | 11.0% | 72 | 14.4% |
| | Unemployed | 174 | 34.8% | 95 | 19.0% |
| Health insurance | Insured | 337 | 67.4% | 302 | 60.4% |
| | Not insured | 163 | 32.6% | 198 | 39.6% |
| Socioeconomic status* | Lower SES (1–5) | 447 | 89.4% | 374 | 74.8% |
| | Higher SES (6–10) | 53 | 10.6% | 126 | 25.2% |
| Family history of T2D | History of T2D | 267 | 53.4% | 205 | 41.0% |
| | No history of T2D | 233 | 46.6% | 295 | 59.0% |
| Time living with T2D | <6 years | 191 | 38.2% | 166 | 33.2% |
| | ≥6 years | 309 | 61.8% | 334 | 66.8% |
| Mean number of comorbidities (SD) | | 500 | 3.4(3.9) | 368 | 1.5(0.8) |

Notes: urban locations were Nairobi and Kiambu (Kenya) and Dar es Salaam (Tanzania); rural/ peri-urban locations were Nyeri and Vihiga (Kenya) and Morogoro (Tanzania).
*As measured by a self-report ladder scale where one represents low SES and 10 high SES.
SES, socioeconomic status.

using wider evidence, such as the international response to COVID-19, to inform their decision to get vaccinated:

I feel like what the government is saying, other countries are also doing it, I don't see how it can be a bad thing [Female-220312_1439-rural-Kenya].

In addition to formal sources of information, people in both countries also used informal local knowledge, which was not always helpful, to inform their decision-making. Interviewees, particularly in rural Tanzania, described how myths and misconceptions about the COVID-19

vaccine in their communities increased vaccine hesitancy locally:

People say if you vaccinate you can become vampire-like and later you will be like a zombie! You can even give birth to an abnormal child [Female-PT12-rural Tanzania].

These myths and misconceptions were fuelled by the speed of the development and rollout of COVID-19 vaccines. Interviewees in both countries expressed concerns about vaccine quality and safety, including the

**Table 2** Multivariate logistic regression results of potential predictors of COVID-19 vaccine uptake

| Variable | Description | Kenya | | Tanzania | |
|---|---|---|---|---|---|
| | | AOR (p value) | (95% CI) | AOR (p value) | (95% CI) |
| Place of residence | Urban (ref) | 1.00 | | 1.00 | |
| | Rural | 1.09 (0.789) | (0.59 to 2.00) | 0.86 (0.605) | (0.49 to 1.50) |
| Sex | Female (ref) | 1.00 | | 1.00 | |
| | Male | 1.11 (0.696) | (0.65 to 1.92) | 1.30 (0.309) | (0.78 to 2.17) |
| Marital status | Not married (ref) | 1.00 | | 1.00 | |
| | Married | 1.56 (0.008) | (0.95 to 2.57) | 1.55 (0.086) | (0.94 to 2.57) |
| Age group | <40 years (ref) | 1.00 | | 1.00 | |
| | 40–49 years | 1.44 (0.402) | (0.61 to 3.37) | 1.21 (0.761) | (0.35 to 4.19) |
| | 50–59 years | 2.51 (0.029) | (1.10 to 5.72) | 1.56 (0.438) | (0.51 to 4.76) |
| | 60–69 years | 2.48 (0.038) | (1.05 to 5.84) | 2.11 (0.200) | (0.67 to 6.61) |
| | > 70 years | 2.46 (0.078) | (0.90 to 6.71) | 2.19 (0.270) | (0.54 to 8.82) |
| Education level | No education (ref) | 1.00 | | 1.00 | |
| | Primary education | 1.38 (0.596) | (0.42 to 4.57) | 1.47 (0.510) | (0.47 to 4.56) |
| | Secondary education | 1.54 (0.490) | (0.45 to 5.21) | 1.79 (0.345) | (0.53 to 6.03) |
| | Higher (college/university) education | 4.25 (0.049) | (1.00 to 18.03) | 4.07 (0.046) | (1.03 to 16.12) |
| Religion | Catholic (ref) | 1.00 | | 1.00 | |
| | Protestants | 0.87 (0.620) | (0.49 to 1.53) | 1.19 (0.565) | (0.66 to 2.13) |
| | Muslims | 0.67 (0.512) | [0.20 to 2.23] | 0.94 (0.822) | (0.55 to 1.62) |
| Occupation status | Formal sector workers (ref) | 1.00 | | 1.00 | |
| | Farmers | 2.86 (0.115) | (0.77 to 10.62) | 1.15 (0.777) | (0.44 to 3.00) |
| | Self-employed business | 1.33 (0.623) | (0.43 to 4.14) | 1.16 (0.742) | (0.47 to 2.87) |
| | Taking care of home | 1.05 (0.952) | (0.20 to 5.58) | | |
| | Retired | 1.26 (0.735) | (0.33 to 4.91) | 0.89 (0.809) | (0.34 to 2.32) |
| | Unemployed | 0.81 (0.728) | (0.26 to 2.58) | 0.99 (0.991) | (0.35 to 2.80) |
| Health insurance | Insured | 1.70 (0.026) | (1.07 to 2.70) | 1.81 (0.035) | (1.04 to 3.13) |
| | Not insured (ref) | 1.00 | | 1.00 | |
| Socioeconomic status (ladder scale 1–10) | Lower SES (1–5) | 0.79 (0.522) | (0.38 to 1.63) | 0.62 (0.059) | (0.37 to 1.02) |
| | Higher SES (6–10) (ref) | 1.00 | | 1.00 | |
| Family history of T2D | Yes | 1.42 (0.123) | (0.91 to 2.21) | 1.35 (0.189) | (0.86 to 2.10) |
| | No (reference) | 1.00 | | 1.00 | |
| Time living with T2D | <6 years | 0.75 (0.228) | (0.47 to 1.20) | 0.75 (0.249) | (0.46 to 1.22) |
| | ≥6 years (ref) | 1.00 | | 1.00 | |
| Comorbidities | Number of comorbidities | 1.08 (0.691) | (0.75 to 1.55) | 1.28 (0.042) | (1.01 to 1.61) |

AOR, adjusted OR; SES, socioeconomic status.

potential of adverse side effects. For example, one man from rural Kenya described how the fear was intensified for many people with T2D:

I think even now most diabetic people are afraid of getting the vaccine. I was also afraid at first because it was believed that if you get vaccinated and you have diabetes you will die [Male-220312_1220-rural-Kenya].

Health insurance emerged as another strong predictor of vaccine uptake in the quantitative analysis: people with health insurance were almost twice as likely to be vaccinated than those who were not insured in Kenya (AOR=1.70, 95% CI 1.07 to 2.70, p<0.05) and Tanzania (AOR=1.81, 95% CI 1.04 to 3.13, p<0.05) (table 2). The interviews suggested that the lower vaccine uptake among participants without health insurance may in part reflect the fact that the financial hardship caused by the pandemic meant that accessing healthcare for those who were not insured was less important than other competing priorities:

I run a small business, so, whatever I get I put it aside to feed my children, and some to buy my diabetic treatment because I don't have health insurance. If I had

the insurance, it would have been easier, I could just come to the hospital and I will be given medication. But now I need to buy (medicine) and the economic situation isn't good with this CORONA at hand. If I had the money, I would buy medicines to take me for two weeks or a month, and I would not miss any dose, but now I can't [Female PT03-urban-Tanzania].

In Kenya, older patients (50–59 years and 60–69 years) had a higher odds of vaccine uptake (AOR=2.51, 95% CI 1.10 to 5.72, p<0.05) and (AOR=2.48, 95% CI 1.05 to 5.84, p<0.05), respectively, compared with young patients (<40 years) (table 2). In Tanzania (but not Kenya), having more comorbidities was associated with higher vaccine uptake (AOR=1.28, 1.01 to 1.61, p<0.05). Being married showed a weak positive association with vaccine uptake in both countries (p<0.10), and lower SES was weakly associated with lower vaccine uptake in Tanzania only (p=0.059). The Modified Poisson regression results (online supplemental appendix 1) were broadly similar to the logistic regression results in terms of associations and levels of statistical significance of both ORs and risk ratios. However, the associations between higher education and being married and vaccine uptake disappeared in Kenya. It is important to note that the results between two models are not directly comparable because the logistic model used OR while Modified Poisson model used risk ratios.

## DISCUSSION

Our investigation of COVID-19 vaccine uptake among people with T2D found that 75% and 26% reported having had at least one vaccination in Kenya and Tanzania, respectively. Vaccine uptake was lower in Tanzania partly because of the vaccine hesitancy of the past presidential regime by arguing to conduct a robust evaluation before accepting the use of vaccines,[31–33] which continued to influence people's decision-making around whether to be vaccinated, despite the subsequent government's promotion of vaccination during the national rollout from July 2021.[26] Being better educated and having health insurance were significantly associated with higher COVID-19 vaccine uptake in both countries. Vaccine uptake was also higher among older people with T2D in Kenya, and among those with increased number of comorbidities in Tanzania. Interviewees reported that concerns about becoming severely ill if they contracted COVID-19 informed their decision to get vaccinated, while fear of vaccine side effects/safety/quality, different and sometimes conflicting information on vaccines, and local myths and misinformation limited vaccine uptake.

The finding that higher education levels were associated with increased vaccine uptake in both countries (although in Kenya the association disappeared in the Poisson sensitivity analysis) is consistent with previous research on COVID-19 vaccination among people with chronic illness including diabetes in Australia[34] and

Italy,[13] and with a study showing an association between better education and intention to get the COVID-19 vaccine among chronically ill people in Saudi Arabia.[15] However, a similar study in Sudan showed no association between education and vaccine uptake.[9]

The finding that having health insurance was associated with increased COVID-19 vaccine uptake among people with T2D in both countries is consistent with the results of a study of people with chronic conditions who had health insurance in Ethiopia.[16] Evidence suggests that having health insurance improves people's access to healthcare services,[35] reduces incidences of catastrophic health spending,[36] which may help people with health insurance feel more empowered to access vaccines than those without health insurance.

The fact that in Tanzania people with more comorbidities were more likely to have been vaccinated likely reflects their perception of increased vulnerability from contracting COVID-19. This interpretation of the quantitative findings is supported by the qualitative accounts that COVID-19 vaccination was thought to offer protection from severe illness. The findings may also reflect the success of the Tanzanian government's strategy to sensitise and prioritise vulnerable population groups during the July 2021 COVID-19 vaccine rollout. However, the number of comorbidities was not associated with COVID-19 vaccine uptake in Kenya, which may reflect the higher overall vaccination coverage.

In our study, being older was associated with higher COVID-19 vaccine uptake in Kenya, although a weaker association in the Poisson sensitivity analysis. This finding is consistent with a study conducted across five sub-Saharan African countries (Burkina Faso, Ethiopia, Ghana, Nigeria and Tanzania), which reported that vaccine hesitancy was extremely high among young people (in this case adolescents); including 14% in rural Kersa, 23% in rural Ibadan, 31% in rural Nouna, 32% in urban Ouagadougou, 37% in urban Addis Ababa, 48% in rural Kintampo, 65% in urban Lagos, 76% in urban Dar es Salaam, and 88% in rural Dodoma.[37] However, other studies have shown mixed associations between vaccine uptake and age among people with T2D. For instance, age was positively associated with COVID-19 vaccination in Australia,[34] but negatively associated in Saudi Arabia.[38] Higher SES was also weakly associated with increased vaccine uptake in Tanzania, which supports the results of previous studies that included people with T2D in Australia,[34] China[8] and Ethiopia.[16]

The fear of vaccine side effects, safety and quality emerged among factors limiting the uptake of COVID-19 vaccination in both countries. These findings are consistent with other literature about COVID-19 vaccine hesitancy in LMICs,[6–10 33] which have highlighted particular concerns about death, developing new diseases, and infertility. Such concerns have been heightened by the rapid development and production of COVID-19 vaccines and the novel mRNA-based vaccine technology.[39]

To our knowledge, this is the first large population-based, mixed-methods study to examine COVID-19 vaccine uptake, as well as the factors influencing uptake, among people with T2D in two LMICs with different national approaches to combatting COVID-19. The study offers insights into the impact on vaccine uptake of these different approaches, which included initial political hesitancy towards vaccination in Tanzania which delayed the vaccination rollout, compared with a pro-vaccine view within the Kenyan government. It is also the first study to examine the influence of comorbidities on COVID-19 vaccine uptake in people with T2D.

Our study had some limitations. First, we were unable to get a fully representative sample of people with T2D in each country, because we only recruited those who were diagnosed and had attended care in health facilities. Vaccine uptake may be higher in this subgroup because they are more engaged with health services than those who do not attend local health facilities. Future studies should recruit people with T2D in community settings to include those who are undiagnosed and/or not registered at health facilities. Second, our study assessed vaccine uptake when national rollout had not yet been fully implemented, particularly in Tanzania. However, in both Kenya and Tanzania, people with chronic diseases such as T2D were prioritised, so it is likely that our study participants would all have been offered at least one vaccination. Third, as COVID-19 vaccination status was self-reported, social desirability bias may have inflated reports of being vaccinated. Fourth, we were unable to use the past history of COVID-19 infection as a predictor of COVID-19 vaccine uptake, because our data had few participants who confirmed to have contracted COVID-19 infection (eg, only 4% of patients with T2D confirmed to have contracted COVID-19 infection in Tanzania).

Our study has important implications for improving COVID-19 vaccine uptake among people with chronic diseases like T2D in countries like Kenya and Tanzania. It highlights the need for governments to engage with and promote evidence-based health advice (such as the effectiveness of COVID-19 vaccines in reducing disease transmission and severity) to encourage vaccine uptake. It also provides a basis to help policy-makers in Kenya and Tanzania develop clear policies and strategies to reduce vaccine hesitancy among people with T2D and other chronic diseases. Potential strategies for consideration include national COVID-19 vaccination education campaigns targeting different 'vaccine hesitant' subgroups, such as those who are less well educated (including low SES), those who do not have health insurance and younger people. However, it is also important to counter the antivaccination views, perceptions and beliefs that can rapidly undermine efforts to promote vaccine uptake,[40] for example, by using influential leaders and role models in local and national multimedia communication campaigns.

**Contributors** SMM, GA, FSM and CMG designed the study. PB and CHK analysed the survey data with support from GA. IM, SMM and SFM analysed qualitative data from both countries. SMM, CMG, FSM and GA contributed to data interpretation. PB and SMM wrote jointly the first draft of the manuscript. CMG, FSM and GA supported development of subsequent drafts. SMM is responsible for the overall content as the guarantor. All authors approved the final manuscript.

**Funding** Medical Research Council (MRC) and the National Institute for Health Research (NIHR) [MR/V035924/1].

**Competing interests** None declared.

**Patient and public involvement** Patients and/or the public were not involved in the design, or conduct, or reporting, or dissemination plans of this research.

**Patient consent for publication** Not applicable.

**Ethics approval** Ethical approval was obtained from respective ethics committees in Kenya and Tanzania. In Kenya, the permit was obtained from the African Population and Health Research Center Ethics Review Committee (ref. no: DOR 2021/041), African Medical and Research Foundation (AMREF) Health Africa's Ethics and Scientific Research Committee (ref. no: AMREF-ESRC P900/2020), and National Commission for Science, Technology and Innovation (licence no: NACOSTI/ p22/14986). In Tanzania, the institutional ethical approval was from the Ifakara Health Institute (IHI/IRB/No: 38-2021), while the national approval was from the National Institute of Medical Research (NIMR/HQ/R.8a/Vol.X/3806). We sought written informed consent from all respondents by providing information on the objectives of the research and the procedures applied in a clear language using an information sheet.

**Provenance and peer review** Not commissioned; externally peer reviewed.

**Data availability statement** Data are available upon reasonable request. Data is available on request. There are ethical restrictions on sharing the data set publicly, one reason being data containing sensitive patient information particularly the experience with COVID-19. The restriction was imposed by the Ifakara Health Institute (IHI) Institutional Review Board (IRB). In case of data request, contact the IRB secretary, Dr. Mwifadhi Mrisho (+255655766675, mmrisho@ihi.or.tz).

**ORCID iDs**
Peter Binyaruka http://orcid.org/0000-0002-1892-7985
Caroline H Karugu http://orcid.org/0000-0002-6914-9714
Shukri F Mohamed http://orcid.org/0000-0002-8693-1943
Frances S Mair http://orcid.org/0000-0001-9780-1135
Cindy M Gray http://orcid.org/0000-0002-4295-6110

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
