## [Reviewer comments · BMJ Open]

ARTICLE DETAILS

TITLE (PROVISIONAL)	Factors associated with COVID-19 vaccine uptake among people with type 2 diabetes in Kenya and Tanzania: a mixed-methods study
AUTHORS	Binyaruka, Peter; Mtenga, Sally M.; Mashasi, Irene; Karugu, Caroline; Mohamed, Shukri; Asiki, G; Mair, Frances; Gray, Cindy

VERSION 1 – REVIEW

REVIEWER	Weerahandi, Himali UCSF
REVIEW RETURNED	29-Apr-2023

GENERAL COMMENTS	Thank you for the opportunity to review this manuscript. This is an interesting and overall well-written manuscript on a large mixed-methods study examining factors influencing COVID-19 vaccine uptake in people with type II diabetes in Kenya and Tanzania. They find that COVID-19 vaccine uptake was lower in Tanzania compared to Kenya and identified patient level characteristics associated with vaccination, which were qualified further with interviews. I think the major takeaway from this study is that it demonstrates what many have suspected---that a country's political leadership has a great influence on the population's uptake on public health interventions such as vaccination. I have some questions and comments that I hope will improve the clarity of the manuscript: Methods: 1. For the study design, the authors state that they surveyed 500 people with Type II DM in each country. How did you come up with this number? Was a power calculation done given the multivariable logistic regression performed?2. Similarly, 21 qualitative interviews were done in Kenya and 30 were done in Tanzania. Usually in studies reporting qualitative work, there is a statement that enrollment was stopped after reaching theoretical saturation. Was that the case here?3. Please provide more details about how study participants were identified and recruited. The authors state that the survey included adults with T2D before COVID-19 in different regions of Kenya and Tanzania and "were identified from health facility outpatient registers and approached for written informed consent." Which facilities were these? Did you approach people in person who happened to have an appointment at these facilities on the days study staff were able to recruit participants in person? The data collection sounds like it was completed within 2 months (page 7, lines 7-8, "Data collection
---

	was conducted in February-March 2022. In Tanzania, all data collection was conducted face-to-face”) which is very fast for 500 participants, particularly when it includes 51 in-depth interviews. 4. Please provide more information about the trained fieldworkers who conducted the surveys and interviews. Were the fieldworkers recruited from the respective countries (Kenya and Tanzania)? Was there always language concordance between the interviewers and interviewees or were translators sometimes used? 5. The surveys in Tanzania were conducted in person---where did the surveys occur? In the clinic? Did this happen in the waiting room or in a private room? I’m wondering because the level of privacy the respondent has during the survey may affect their responses. 6. Measures: please provide more information about the socioeconomic status (SES) scale self-report ladder scale. How does a person determine what number they are? Results 7. Having college/university education was associated with vaccine uptake in both countries in the primary multivariable logistic regression model, but the effect went away in the Poisson model. Why do think that is? Also, the effect of age also went away in the Poisson model for Kenya. Do you think this is because the model was underpowered? Also, given these borderline findings, I think the significance of these findings should be de-emphasized in the discussion. Currently the discussion states on page 10, line 49 that “Being better educated...[was] significantly associated with higher COVID-19 vaccine uptake in both countries.” And on page 11 line 3 “the finding that higher education levels were associated with increased vaccine uptake...” Similarly, the paragraph in the discussion about “being older was associated with higher COVID-19 vaccine uptake in Kenya...” (page 12, line 41) should be qualified that the significance of the findings were borderline given that they didn’t hold up in the sensitivity analyses with the Poisson model. 8. Table 1, Occupation: What does “Formal workers” mean?
--	---

REVIEWER	Venkatesan, Balamurali SRM Medical College Hospital and Research Centre
REVIEW RETURNED	06-May-2023

GENERAL COMMENTS	 1. Mention the prevalence of COVID-19 infection as well as diabetes in both the countries 2. In the "Measures" section, include past history of COVID-19 infection and COVID-19 vaccination. 3. Mention the numerical results of various studies which you have been correlated with your current research results.
---

VERSION 1 – AUTHOR RESPONSE

Reviewer: 1

Dr. Himali Weerahandi, UCSF

Comments to the Author:

Thank you for the opportunity to review this manuscript.

This is an interesting and overall well-written manuscript on a large mixed-methods study examining factors influencing COVID-19 vaccine uptake in people with type II diabetes in Kenya and Tanzania. They find that COVID-19 vaccine uptake was lower in Tanzania compared to Kenya and identified patient level characteristics associated with vaccination, which were qualified further with interviews. I think the major takeaway from this study is that it demonstrates what many have suspected---that a country's political leadership has a great influence on the population's uptake on public health interventions such as vaccination.

Response: Thank you! We agree with your observation regarding a takeaway message.

I have some questions and comments that I hope will improve the clarity of the manuscript:

Methods:

1. For the study design, the authors state that they surveyed 500 people with Type II DM in each country. How did you come up with this number? Was a power calculation done given the multivariable logistic regression performed?

Response: We have added the explanation about sample size calculation (page 5): “The sample size of 500 people was estimated using the Cochran formula, assuming 50% of patients with T2D experienced disruption of care during COVID-19, with 80% power, a 5% margin of error and a 30% non-response rate.”

2. Similarly, 21 qualitative interviews were done in Kenya and 30 were done in Tanzania. Usually in studies reporting qualitative work, there is a statement that enrollment was stopped after reaching theoretical saturation. Was that the case here?

Response: We agree with the reviewer, and a sentence has been added (page 5): “In both countries, data collection stopped once saturation of views was achieved.”

3. Please provide more details about how study participants were identified and recruited. The authors state that the survey included adults with T2D before COVID-19 in different regions of Kenya and Tanzania and “were identified from health facility outpatient registers and approached for written informed consent.” Which facilities were these? Did you approach people in person who happened to have an appointment at these facilities on the days study staff were able to recruit participants in person? The data collection sounds like it was completed within 2 months (page 7, lines 7-8, “Data collection was conducted in February-March 2022. In Tanzania, all data collection was conducted face-to-face”) which is very fast for 500 participants, particularly when it includes 51 in-depth interviews.

Response: We have expanded the paragraph on study participants by including the following sentence (page 5): *Adult patients with T2D who received care in selected hospitals and health centres before March 2020 were identified from health facility outpatient registers and approached for written informed consent.*” It was possible to finish data collection from all 500 patients between February and March 2022 because we recruited a large number of data collectors.

4. Please provide more information about the trained fieldworkers who conducted the surveys and interviews. Were the fieldworkers recruited from the respective countries (Kenya and Tanzania)? Was there always language concordance between the interviewers and interviewees or were translators sometimes used?

Response: We have added a sentence (page 5) “Fieldworkers were recruited from each country and trained for four days before data collection.”

In terms of language, we have added a sentence (page 5): “Data collection was conducted in Swahili in Tanzania, while in Kenya, where fieldworkers were fluent in Swahili and English, participants were allowed to respond in either language.”

5. The surveys in Tanzania were conducted in person---where did the surveys occur? In the clinic? Did this happen in the waiting room or in a private room? I'm wondering because the level of privacy the respondent has during the survey may affect their responses.

Response: We have added additional explanation (page 5): “In Tanzania, all data collection was conducted face-to-face at the health facility in a room or other location that ensured full privacy for the participant.”

6. Measures: please provide more information about the socioeconomic status (SES) scale self-report ladder scale. How does a person determine what number they are?

Response: We have expanded with additional explanation under the “measures” section (page 6). “... socioeconomic status (SES) based on a ladder scale (1-10) that asked participants to rank their households in terms of economic position, where 1 was lowest economic position and 10 was highest economic position (lower 1-5/higher 6-10).”

Results

7. Having college/university education was associated with vaccine uptake in both countries in the primary multivariable logistic regression model, but the effect went away in the Poisson model. Why do you think that is? Also, the effect of age also went away in the Poisson model for Kenya. Do you think this is because the model was underpowered? Also, given these borderline findings, I think the significance of these findings should be de-emphasized in the discussion. Currently the discussion states on page 10, line 49 that “Being better educated...[was] significantly associated with higher COVID-19 vaccine uptake in both countries.” And on page 11 line 3 “the finding that higher education levels were associated with increased vaccine uptake...” Similarly, the paragraph in the discussion about “being older was associated with higher COVID-19 vaccine uptake in Kenya...” (page 12, line 41) should be qualified that the significance of the findings were borderline given that they didn’t hold up in the sensitivity analyses with the Poisson model.

Response: We are aware that some predictors were no longer statistically significant in the Poisson model. This is mainly due to differences in estimations between two models – they use similar explanatory variables but different measure of outcomes and ways of presenting results. For instance, logistic model presented result in Odds Ratio, while modified Poisson presented result in Risk Ratio. We have added a sentence when explaining the reduced level of statistical significance in a modified Poisson model (page 10). “It is important to note that the results between two models are not directly comparable because the logistic model used odds ratio while Modified Poisson model used risk ratios.”

We have also edited the discussion to de-emphasize the weak association by specifying some weak predictors like education (page 10), and age in Kenya (page 11).

Page 10 “The finding that higher education levels were associated with increased vaccine uptake in both countries (although in Kenya the association disappeared in the Poisson sensitivity analysis)”

Page 11 “In our study, being older was associated with higher COVID-19 vaccine uptake in Kenya, albeit a weaker association in the Poisson sensitivity analysis.”

8. Table 1, Occupation: What does “Formal workers” mean?

Response: We have edited in Table 1 and 2. It was meant for formal sector workers (including public and private sector).

Reviewer: 2

**Dr. Balamurali Venkatesan, SRM Medical College Hospital and Research Centre
Comments to the Author:**

1. Mention the prevalence of COVID-19 infection as well as diabetes in both the countries

Response: For the prevalence of COVID-19 infection, we have added a sentence (page 4): “By August 2023, Tanzania and Kenya had recorded 43,078 and 343,918 confirmed cases and 846 and 5,689 COVID-19-related deaths, respectively.” (Cited from a webpage link for the WHO Health Emergency Dashboard <https://covid19.who.int>)

2. In the "Measures" section, include past history of COVID-19 infection and COVID-19 vaccination.

Response: This would have been an interesting predictor of COVID-19 uptake. Unfortunately, our datasets did not have enough observations of COVID-19 infections to be added as predictor. For example, only 4% with T2D confirmed to have contracted COVID-19 in

Tanzania. Knowing this caveat, we have added a fourth study limitation (page 12): ***“Fourth, we were unable to use the past history of COVID-19 infection as a predictor of COVID-19 vaccine uptake, because our data had few participants who confirmed to have contracted COVID-19 infection (e.g., only 4% of patients with T2D confirmed to have contracted COVID-19 infection in Tanzania).”***

3. Mention the numerical results of various studies which you have been correlated with your current research results.

Response: We have inserted some numerical percentage in the discussion particularly on the discussion about age. (Page 11): ***“This finding is consistent with a study conducted across five sub-Saharan African countries (Burkina Faso, Ethiopia, Ghana, Nigeria, and Tanzania), which reported that vaccine hesitancy was extremely high among young people (in this case adolescents); including 14% in rural Kersa, 23% in rural Ibadan, 31% in rural Nouna, 32% in urban Ouagadougou, 37% in urban Addis Ababa, 48% in rural Kintampo, 65% in urban Lagos, 76% in urban Dar es Salaam, and 88% in rural Dodoma.³⁵”***